# The E3 Ubiquitin-Protein Ligase RNF4 Promotes TNF-α-Induced Cell Death Triggered by RIPK1

**DOI:** 10.3390/ijms22115796

**Published:** 2021-05-28

**Authors:** Tatsuya Shimada, Yuki Kudoh, Takuya Noguchi, Tomohiro Kagi, Midori Suzuki, Mei Tsuchida, Hiromu Komatsu, Miki Takahashi, Yusuke Hirata, Atsushi Matsuzawa

**Affiliations:** Laboratory of Health Chemistry, Graduate School of Pharmaceutical Sciences, Tohoku University, 6-3 Aoba, Aramaki, Aoba-ku, Sendai 980-8578, Japan; tatsuya.shimada.t7@dc.tohoku.ac.jp (T.S.); yuki1127yb@gmail.com (Y.K.); toho131@dc.tohoku.ac.jp (T.K.); midori.suzuki.r5@dc.tohoku.ac.jp (M.S.); mei.tsuchida.s3@alumni.tohoku.ac.jp (M.T.); hiromu.komatsu.p1@dc.tohoku.ac.jp (H.K.); miki.takahashi.r1@gmail.com (M.T.); yusuke.hirata.d8@tohoku.ac.jp (Y.H.)

**Keywords:** receptor-interacting protein kinase 1 (RIPK1), Tumor necrosis factor-α (TNF-α), RING finger protein 4 (RNF4), TNF receptor-mediated cell death

## Abstract

Receptor-interacting protein kinase 1 (RIPK1) is a key component of the tumor necrosis factor (TNF) receptor signaling complex that regulates both pro- and anti-apoptotic signaling. The reciprocal functions of RIPK1 in TNF signaling are determined by the state of the posttranslational modifications (PTMs) of RIPK1. However, the underlying mechanisms associated with the PTMs of RIPK1 are unclear. In this study, we found that RING finger protein 4 (RNF4), a RING finger E3 ubiquitin ligase, is required for the RIPK1 autophosphorylation and subsequent cell death. It has been reported that RNF4 negatively regulates TNF-α-induced activation of the nuclear factor-κB (NF-κB) through downregulation of transforming growth factor β-activated kinase 1 (TAK1) activity, indicating the possibility that RNF4-mediated TAK1 suppression results in enhanced sensitivity to cell death. However, interestingly, RNF4 was needed to induce RIPK1-mediated cell death even in the absence of TAK1, suggesting that RNF4 can promote RIPK1-mediated cell death without suppressing the TAK1 activity. Thus, these observations reveal the existence of a novel mechanism whereby RNF4 promotes the autophosphorylation of RIPK1, which provides a novel insight into the molecular basis for the PTMs of RIPK1.

## 1. Introduction

Tumor necrosis factor-α (TNF-α) is an inflammatory cytokine that regulates a wide variety of cellular responses, including cell growth, differentiation, immune responses, and cell death [1]. Upon binding TNF-α to TNF receptor 1 (TNFR1), the signaling components are recruited to TNFR1 and form the TNFR1 signaling complex called complex I [2]. Basically, the main components of TNFR1 signaling complex (complex I), such as receptor-interacting protein kinase 1 (RIPK1), tumor necrosis factor receptor-associated factor 2 (TRAF2), and transforming growth factor β-activated kinase 1 (TAK1), elicit activation of the nuclear factor-κB (NF-κB) pathways, and the mitogen-activated protein kinase (MAPK) pathways, such as the c-Jun N-terminal kinase (JNK) and p38 MAPK pathways, that preferentially induce anti-apoptotic and pro-inflammatory responses [3,4,5]. On the other hand, conformational rearrangements of complex I occur in a context-dependent manner, resulting in the formation of the cytosolic complex (called complex II) that functions as a signaling hub to induce cell death, such as apoptosis and necroptosis [6]. Interestingly, posttranslational modifications of RIPK1 play a key role in cell fate decisions in response to TNF-α. On complex I, RIPK1 is polyubiquitinated by its E3 ligases, a cellular inhibitor of apoptosis 1 and 2 (cIAP1/2), and then mediates anti-apoptotic signals and inhibits the formation of complex II. Meanwhile, deubiquitination of RIPK1 allows the formation of complex II, leading to the activation of proapoptotic signals [7]. Moreover, phospho-dependent regulation of RIPK1 controls the induction of cell death [8,9,10]. Autophosphorylation of RIPK1 at serine (Ser) 166 (S166) promotes RIPK1-mediated cell death, whereas phosphorylation at Ser 321 (S321) by TAK1 or MAPKAPK2 (MK2) suppresses it [9,10,11]. Therefore, the posttranslational modifications of RIPK1 appear to function as a signaling switch that determines cellular responses to TNF-α. However, its precise mechanisms are not fully understood.

RING finger protein 4 (RNF4) is a RING finger E3 ubiquitin ligase classified as a small ubiquitin-related modifier (SUMO)-targeted ubiquitin ligase (STUbL) [12,13]. RNF4 recognizes and ubiquitinates polysumoylated proteins via its SUMO-interacting motif (SIM) domain and leads to ubiquitination-dependent proteasome degradation [14]. To date, several target proteins of RNF4 have been identified, including Mediator of DNA damage checkpoint 1 (MDC1), breast cancer susceptibility gene I (BRCA1), and progressive multifocal leukoencephalopathy (PML), many of which are involved in DNA repair, nucleic acid metabolism, and chromatin regulation. Therefore, RNF4 is considered to play important roles in DNA damage response and genome homeostasis [15,16,17,18]. On the other hand, it has been reported that RNF4 modulates inflammatory responses by regulating nuclear receptor 4A 1 (NR4A) and TAK1 signaling [19,20]. RNF4 directly targets NR4A1, a member of the steroid nuclear hormone receptor superfamily that regulates a wide range of inflammatory responses and controls inflammatory cytokine signaling and macrophage cell death [19]. Meanwhile, RNF4 inhibits NF-κB activation induced by TNF-α and interleukin-1 (IL-1β) by promoting lysosomal degradation of TAK1-binding protein 2 (TAB2), an essential subunit of the TAK1 signaling complex that enhances the kinase activity of TAK1 [20]. Thus, RNF4 has emerged as a key mediator of inflammatory signaling.

In this study, we identified RNF4 as a novel regulator of RIPK1 that promotes TNF-α-induced autophosphorylation of RIPK1 at S166 and subsequent cell death. Interestingly, RNF4 promotes TNF-α-induced cell death under conditions where the TAK1-mediated NF-κB activation is blocked. Therefore, these results suggest that RNF4 promotes the autophosphorylation of RIPK1 through a mechanism independent of its ability to suppress the TAK1-mediated NF-κB signaling.

## 2. Results

### 2.1. RNF4 Is Specifically Required for RIPK1-Mediated Cell Death Stimulated by TNF-α

To investigate the role of RNF4 in TNF signaling, we established RNF4 knockout (KO) cells in mouse embryonic fibroblasts (MEFs) by using the clustered regularly interspaced short palindromic repeat (CRISPR)/CRISPR-associated protein 9 (Cas9) system (Figure 1A,B). We then tested whether RNF4 is required for TNF-α-induced cell death. TNF-α induces cell death through at least two distinct mechanisms classified as RIPK1-dependent and -independent pathways [7]. Experimentally, the RIPK1-dependent mechanism is activated by TNF-α in the presence of second mitochondria-derived activator of caspase (Smac) mimetics that act as selective inhibitors of cellular inhibitors of apoptosis proteins (cIAPs), including BV-6 [7,21]. On the other hand, treatment with TNF-α in combination with the protein-synthesis inhibitor cycloheximide (CHX), which blocks TNF-α-mediated induction of anti-apoptotic proteins, activates the RIPK1-independent mechanism [7]. Indeed, the RIPK1 inhibitor Necrostatin-1 (Nec-1) could suppress TNF-α-induced cell death mediated by RIPK1-denendent (co-treatment with BV-6) but not -independent (co-treatment with CHX) mechanism (Figure 1C,D). Interestingly, both clones of RNF4 KO MEFs exhibited significant resistance only when RIPK1-mediated cell death was induced, indicating that RNF4 is selectively required for TNF-α-induced cell death mediated by RIPK1 (Figure 1E,F). We, therefore, investigated the roles of RNF4 in the RIPK1-mediated cell death. As shown in Figure 1G, we found that TNF-α-induced cleavage of caspase-8 to the p18 fragment that has the enzymatic activity, and its downstream caspase-3 (a well-known indicator of TNF-α-induced apoptosis) was clearly abrogated in RNF4 KO MEFs. Moreover, the colorimetric caspase-8 assay confirmed the abrogation in RNF4 KO MEFs, showing that RNF4 acts upstream of caspase-8 and stimulates the proapoptotic process (Figure 1H).

### 2.2. The E3 Ubiquitin Ligase Activity of RNF4 Is Required for TNF-α-Induced Apoptosis

We next examined whether RNF4 promotes TNF-α-induced cell death by exerting its E3 ubiquitin ligase activity. To this end, we established RNF4-reconstituted MEFs (Figure 2A). As we expected, the reconstitution of RNF4 wild-type (WT) in RNF4 KO MEFs successfully restored sensitivity to TNF-α-induced apoptosis to a similar extent as control cells (Figure 2B). On the other hand, RNF4 KO MEFs expressing an enzymatically inactive mutant of RNF4 in which cysteine (Cys) 177 and 180 substituted by Ser (RNF4 CS mutant) showed strong resistance to TNF-α-induced apoptosis when compared with RNF4 WT reconstituted MEFs, even though the expression levels of the RNF4 CS mutant were higher than that of RNF4 WT (Figure 2A,B) [22]. Consistent with this observation, TNF-α-induced caspase-8 activation was more effectively recovered by the reconstitution of RNF4 WT than the CS mutant in both immunoblot and colorimetric caspase-8 assay (Figure 2C,D). These results, therefore, suggest that the E3 ubiquitin ligase activity of RNF4 is required for RNF4-mediated cell death.

### 2.3. RNF4 Suppresses TNF-α-Induced Activation of the NF-κB and MAPK Signaling Pathways

A previous report demonstrated that RNF4 negatively regulates the TAK1-dependent signals, including the NF-κB and MAPK pathways, by downregulating TAB2 [20]. Indeed, TNF-α-induced nuclear translocation of p65 NF-κB, an indicator of the NF-κB activation, was enhanced in RNF4 KO MEFs when compared with WT MEFs (Figure 3A). Moreover, TNF-α-induced activation of MAP kinases, such as p38, JNK, and extracellular signal-regulated kinase (ERK), was also enhanced (Figure 3B). These observations show that RNF4 suppresses the NF-κB and MAPK signaling pathways through the negative regulation of TAK1. 

### 2.4. RNF4 Promotes TNF-α-Induced Cell Death Independently of Its Inhibitory Effects on the TAK1 Signaling

Since the signaling pathways activated by TAK1 basically mediate anti-apoptotic responses, it is known that TAK1 KO cells are sensitive to TNF-α-induced cell death [4]. In order to confirm this observation and to determine the mechanisms by which RNF4 promotes TNF-α-induced cell death, we established TAK1 KO cells in MEFs and human fibrosarcoma HT1080 cells. However, TAK1 KO MEFs (clone #1) still expressed the TAK1 protein due to heterozygous deletion (Figure 4A,B). As shown in Figure 4C,D, both TAK1 KO MEFs and human fibrosarcoma HT1080 cells were predictably sensitive to TNF-α-induced cell death mediated by RIPK1. Therefore, it is possible that RNF4-mediated TAK1 suppression results in enhanced sensitivity to cell death. To explore whether RNF4 promotes TNF-α-induced cell death through only the negative regulation of TAK1, we tested the sensitivity of RNF4 KO MEFs to TNF-α-induced cell death in the presence of the TAK1 kinase inhibitor 5Z-7-oxozeaenol (5Z-7). As shown in Figure 4E, 5Z-7 increased the sensitivity to TNF-α-induced cell death in both WT and RNF4 KO MEFs. However, interestingly, RNF4 KO MEFs still exhibited significant resistance to TNF-α-induced cell death even in the presence of 5Z-7 (Figure 4E). Moreover, knockdown of RNF4 in TAK1 KO HT1080 cells reduced sensitivity to TNF-α-induced cell death similar to that in WT HT1080 cells (Figure 4F,G). Collectively, these results suggest that RNF4 promotes TNF-α-induced cell death through a mechanism independent of its inhibitory effects on the TAK1 signaling.

### 2.5. RNF4 Promotes Phosphorylation of RIPK1 at Ser166

Finally, we examined the effect of RNF4 on the phosphorylation of RIPK1 at Ser166, an essential process triggering RIPK1-mediated cell death. Interestingly, we found that the autophosphorylation of RIPK1 at Ser166 triggered by TNF-α was clearly attenuated in RNF4 KO MEFs at earlier time points than that of the caspase-8 activation (Figure 5A). The attenuation was observed at late time points (Figure 5B). Moreover, the reconstitution of RNF4 WT but not the CS mutant clearly recovered the RIPK1 phosphorylation, suggesting that RNF4 promotes the RIPK1 autophosphorylation in an E3 activity-dependent manner, leading to TNF-α-induced cell death at all time points that we tested (Figure 5C,D).

## 3. Discussion

In the present study, we provide evidence that RNF4 promotes TNF-α-induced cell death by positively regulating the phosphorylation of RIPK1 at Ser166. Additionally, we demonstrate that its ability to promote cell death is exerted in the absence of TAK1. Moreover, as shown in Figure 2B, MEFs expressing the enzymatically inactive mutant of RNF4 (RNF4 CS mutant) exhibited strong resistance to TNF-α-induced cell death. We thus concluded that RNF4 activates RIPK1 through a mechanism distinct from the negative regulation of the TAK1 signaling, which requires the RNF4 E3 ligase activity (Figure 4F). Since RNF4 acts as a STUbL, its target protein may be SUMOylated [12,13]. Moreover, considering that RNF4 caused ubiquitination and proteasomal degradation of its target proteins, negative regulators of RIPK1 are potential candidates of the RNF4 targets. MAP kinase-activated protein kinase 2 (MK2), Ser/Thr kinase unc-51-like kinase 1 (ULK1), TANK-binding kinase 1 (TBK1), and IκB kinase-ε (IKKε) are already known to be the negative regulators of RIPK1 [11,23,24,25]. In particular, TBK1 kinase activity was modulated by SUMOylation, increasing the possibility that RNF4 targets TBK1. On the other hand, RNF4 has been reported to mediate atypical ubiquitination that modulates activity or stability of its target proteins, and hence, has the potential to regulates the RIPK1 kinase activity via noncanonical mechanisms [26,27]. Moreover, RNF4 controls the dynamics of stress granules (SGs) [28]. Interestingly, a recent report revealed that RIPK1 aggregates in SGs, where the downstream signaling is activated [28]. Therefore, it is likely that RNF4 regulates the RIPK1 kinase activity through the formation of the SGs. In any case, although further studies are needed to elucidate the mechanisms by which RNF4 promotes the phosphorylation of RIPK1, our results demonstrated a novel function of RNF4 as a proapoptotic factor in the TNFR signaling.

Accumulating evidence indicates that RNF4 has potential relevance to tumorigenesis [26,29]. RNF4 promotes the activity of oncogenic transcription factors through ubiquitination-dependent stabilization [26]. In some of the cases, both mRNA and protein levels of RNF4 are elevated in human cancer cells, which potentiates the tumorigenic properties of cancer cells, and correlates with poor prognosis and with resistance to anticancer agents [26,29]. Thus, inhibition of RNF4 appears to be a novel anticancer strategy that leads to degradation of the oncoproteins. Meanwhile, RNF4 acts as a tumor suppressor in promyelocytic leukemia (PML) through the degradation of an oncogenic fusion protein between PML protein and retinoic acid receptor α [30]. Since both apoptosis and necroptosis mediated by RIPK1 have been explored as a strategy for anticancer therapy, the ability of RNF4 to promote RIPK1-mediated cell death may contribute to antitumor responses. Ref. [31]. Therefore, elucidation of the mechanisms by which RNF4 promotes RIPK1-mediated cell death may open up possibilities for new therapeutic strategies for cancers associated with overexpression of RNF4.

## 4. Materials and Methods

### 4.1. Cell Culture and Reagents

Human fibrosarcoma cell line HT1080 and Mouse Embryonic Fibroblasts (MEFs) cells were grown in Dulbecco’s Modified Eagle Medium (DMEM), 10% heat-inactivated fetal bovine serum (FBS), and 1% penicillin–streptomycin solution, at 37 °C under a 5% CO_2_ atmosphere. siRNAs were purchased from Qiagen (Hilden, Germany) (RNF4 #1: SI03228512, RNF4 #2: SI04167828). AllStars negative control siRNA (Qiagen) was used as a control. siRNAs were transfected using Lipofectamine RNAiMAX (Merck Millipore, Burlington, VT, USA), according to the manufacturer’s protocol. All reagents were obtained from commercial sources; TNF-α (Enzo Life Sciences, Farmingdale, NY, USA), 5z-7-oxozeaenol(5Z-7), BV-6, necrostatin-1 (Santa Cruz, Dallas, TX, USA), cycloheximide (Sigma, St. Louis, MO, USA). The antibodies used were against caspase-8 (Enzo Life Sciences), RNF4 (Proteintech, Rosemond, CA, USA), FLAG (Sigma), α-tubulin, p65, Fibrillarin (Santa Cruz), caspase-3, phospho-p38, total p38, phospho-JNK, total JNK, phospho-ERK, total ERK, phospho-RIPK1 (Cell signaling, Danvers, MA, USA), total RIPK1 (Becton and Dickinson, Franklin Lakes, NJ, USA), and β-actin (Wako, Tokyo, Japan).

### 4.2. PMS/MTS Assay

PMS/MTS assay was performed to determine cell viability as described previously [32]. Cells were seeded on 96-well plates. After indicated stimulation, cell viability was evaluated by using Cell Titer 96 Cell Proliferation Assay (Promega, Madison, WI, USA), according to the manufacturer’s protocol. The absorbance was read at 492 nm using a microplate reader, and the data are normalized to control (100%) without stimulus.

### 4.3. Immunoblot Analysis

Cells were lysed with the 1% Triton X-100 buffer (20 mM Tris-HCl (pH 7.4), 150 mM NaCl, 1% Triton-X100, 10% Glycerol, and 1% protease inhibitor cocktails (Nacalai Tesque, Kyoto, Japan)). After centrifugation, the cell lysates were separated by SDS-PAGE and evaluated as described previously [33]. The blots were developed with ECL (Merck Millipore). In Figure 3A, nuclear fractions were isolated and subjected to immunoblot analysis as described previously [34].

### 4.4. Generation of Knockout Cell Lines

*TAK1* knockout cells were generated and characterized in the previous study [35]. All KO cells were generated using the CRISPR/Cas9 system as described previously [36]. Guide RNAs (gRNAs) were designed to target a region in the exon 3 of *mRNF4* gene (5′-CACCGCCTCAAAGAAAGCGGCGTGG-3′), that in the exon 5 of *hTAK1* gene (5′-AAACAAGCGCTAATTCACAGGGACC-3′), and that in the exon 5 of *mTAK1* gene (5′-AAACAAGCGCTGATTCACAGGGACC-3′), using CRISPRdirect (https://crispr.dbcls.jp accessed on 1 May 2021) [37]. gRNA-encoding oligonucleotide was cloned into lentiCRISPRv2 plasmid (addgene), and the plasmid was transfected with human embryonic kidney (HEK) 293A cells together with a packaging plasmid psPAX2 and an envelope plasmid pVSV-G. The supernatants were collected and used to infect HT1080 cells or MEFs, and then infected cells were selected with puromycin and cloned by limiting dilution to obtain 100% efficiency. To determine the mutations of *mRNF4*, *hTAK1* and *mTAK1* in cloned cells, genomic sequence around the target region was analyzed by PCR-direct sequencing using extracted DNA from each clone as a template and the following primers: 5′-CCCTTCTTTCTCCCTTGATG-3′ and 5′-CGAAGTTGTTTCCCGAGTTC-3′ for mRNF4; 5′-TTCGGGGTGGTGAGAGTGA-3′ and 5′-TTGTGCCTTTCTTTCGCAGT-3′ for hTAK1. 5′-ACTACACTGCTGCTCATGCC-3′ and 5′-GGGAAGAGAAAAGGGAAAGGC-3′ for mTAK1. 

### 4.5. Colorimetric Caspase-8 Assay

Cells were seeded on 6-well plates. After indicated stimulation, cells were lysed in 50 µL of Cell Lysis Buffer included in the Caspase-8 Colorimetric Assay Kit (Biovision, Milpitas, CA, USA). Ten microliters of soluble cell extracts was mixed with 100 µL of caspase reaction buffer (10 mM Tris-HCl pH7.4, 150 mM NaCl, 0.1% CHAPS, 2 mM MgCl2, 5 mM EGTA, and 1mM DTT) supplemented with the caspase-8-specific substrate IETD-pNA (Biovision) at a final concentration of 100 µM and incubate at 37 ℃ for 1h. Cleaved pNA was measured using a microplate reader with the absorbance was read at 405 nm. Data are normalized to control (100%) without stimulus.

### 4.6. Generation of Reconstituted MEFs

Reconstituted MEFs were generated by retroviral transduction as previously described [38]. A packaging cell line Phoenix-AMPHO was transfected with pMXs-IH inserted with either the RNF4 WT or CS mutant. After 48 h, the growth medium containing retrovirus was collected. MEFs were incubated with the virus-containing medium with 10 μg/mL polybrene for 48 h, and uninfected cells were eliminated by hygromycin selection.

## Figures and Tables

**Figure 1 ijms-22-05796-f001:**
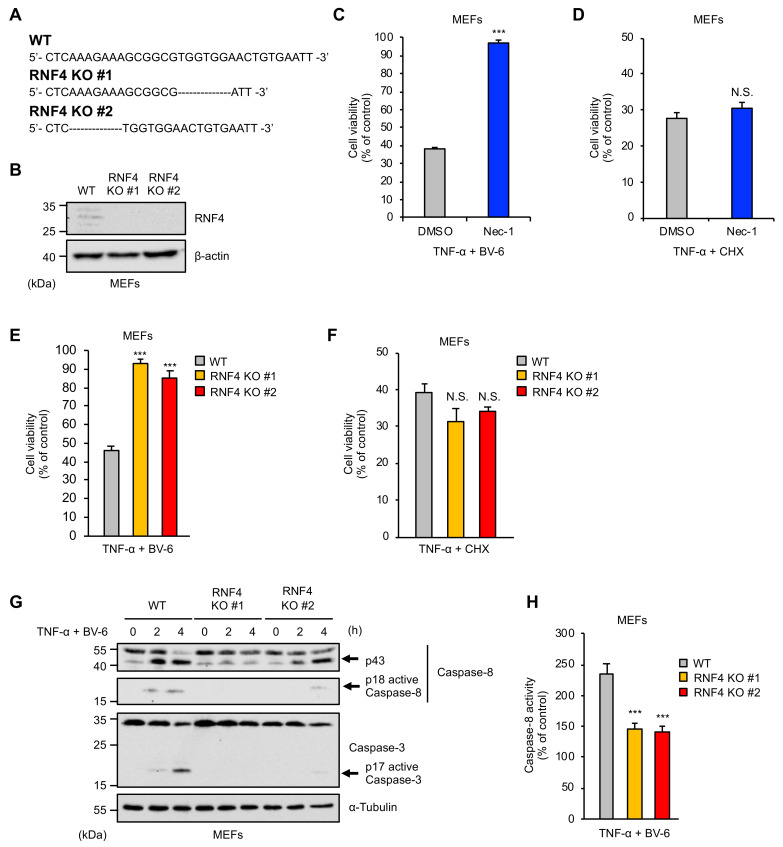
RNF4 is specifically required for RIPK1-mediated cell death stimulated by TNF-α. (**A**) DNA sequences around the guide RNAs (gRNAs) target sites of RNF4. (**B**) Immunoblot analysis of RNF4 in mouse embryonic fibroblasts (MEFs). MEFs were subjected to immunoblotting with the indicated antibodies. β-actin was used as a loading control. (**C**,**D**) Inhibitory effect of RIPK1 kinase inhibitor Nec-1 on TNF-α-induced cell death. (**C**) MEFs were treated with TNF-α (50 ng/mL) for 12 h in the presence of the cIAP inhibitor BV-6 (1 μM), RIPK1 kinase inhibitor Nec-1(10 μM), or Dimethyl Sulfoxide (DMSO) as a control, and then subjected to cell viability assay. (**D**) MEFs were treated with TNF-α (25 ng/mL) for 12 h in the presence of the CHX (15 μg/mL), RIPK1 kinase inhibitor Nec-1(10 μM), or Dimethyl Sulfoxide (DMSO) as a control, and then subjected to cell viability assay. (**E**,**F**) Requirement of RNF4 for TNF-α-induced cell death. (**E**) MEFs were treated with TNF-α (25 ng/mL) for 12 h in the presence of the cIAP inhibitor BV-6 (1 μM) and then subjected to cell viability assay. (**F**) MEFs were treated with TNF-α (25 ng/mL) for 12 h in the presence of the CHX (15 μg/mL) and then subjected to cell viability assay. (**G**,**H**) TNF-α-induced Caspase-8 and Caspase-3 activation in RNF KO MEFs. MEFs were treated with TNF-α (100 ng/mL) for the indicated periods in the presence of BV-6 (1 μM). (**G**) Cell lysates were subjected to immunoblotting with the indicated antibodies. α-Tubulin was used as a loading control. (**H**) MEFs were treated with TNF-α (100 ng/mL) for 6 h in the presence of BV-6 (1 μM). Caspase-8 activity was measured by colorimetric Caspase-8 assay. Data are shown as the ratio of Caspase-8 activity versus corresponding controls. (**C**–**F**) Cell viability was measured by PMS (phenazine methosulfate)/MTS(3-(4,5-dimethylthiazol-2-yl)-5-(3-carboxymethoxyphenyl)-2-(4-sulfophenyl)-2H-tetrazolium) assay. Data shown are the mean ± SD (n = 3). Statistical significance was tested using an unpaired Student’s *t*-test; *** *p* < 0.001, N.S.: not significant. (vs. control cells). All data are representative of at least three independent experiments.

**Figure 2 ijms-22-05796-f002:**
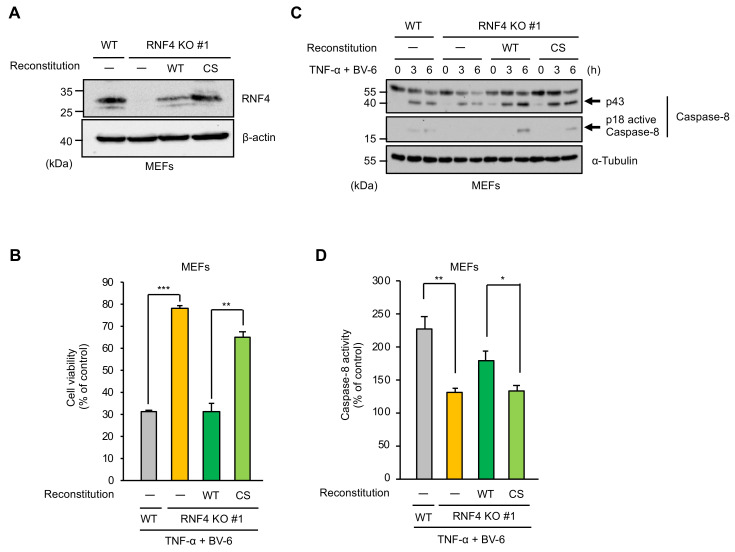
The E3 ubiquitin ligase activity of RNF4 is required for TNF-α-induced apoptosis. (**A**) Immunoblot analysis of RNF4 in MEFs. MEFs were subjected to immunoblotting with the indicated antibodies. β-actin was used as a loading control. (**B**) Effect of the RNF4 reconstitution on TNF-α-induced cell death. MEFs were treated with TNF-α (25 ng/mL) for 12 h in the presence of the cIAP inhibitor BV-6 (1 μM) and then subjected to PMS/MTS assay. Data shown are the mean ± SD (n = 3) Significant differences were assessed by Student’s *t*-test; *** *p* < 0.001, ** *p* < 0.01 (versus control). (**C**,**D**) Effect of the RNF4 reconstitution on TNF-α-induced Caspase-8 activation. (**C**) MEFs were treated with TNF-α (100 ng/mL) for the indicated periods in the presence of BV-6 (1 μM). Cell lysates were subjected to immunoblotting with the indicated antibodies. α-Tubulin was used as a loading control. (**D**) MEFs were treated with TNF-α (100 ng/mL) for 6h in the presence of BV-6 (1 μM). Caspase-8 activity was measured by the Colorimetric Caspase-8 assay. Data are shown as the ratio of Caspase-8 activity versus corresponding controls. Data shown are the mean ± SD (n = 3). Significant differences were assessed by Student’s *t*-test; ** *p* < 0.01, * *p* < 0.05 (versus control). All data are representative of at least three independent experiments.

**Figure 3 ijms-22-05796-f003:**
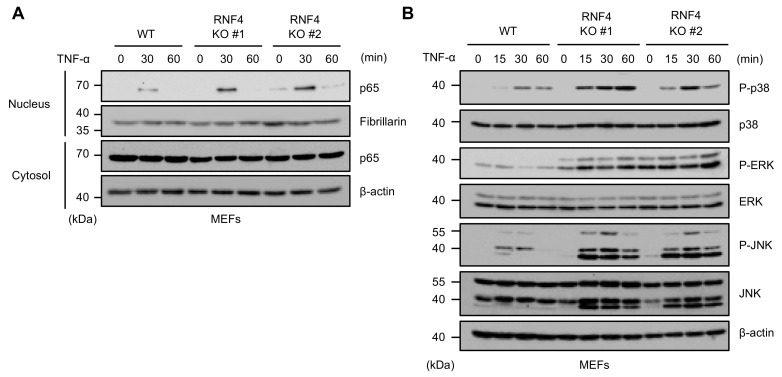
RNF4 suppresses TNF-α-induced activation of the NF-κB and MAPK signaling pathways. (**A**) TNF-α-induced nuclear translocation of p65 in RNF4 KO MEFs. MEFs were treated with TNF-α (50 ng/mL) for the indicated periods. The nuclear and cytoplasmic extracts were subjected to immunoblotting with the indicated antibodies. β-actin (Cytosol) and Fibrillarin (Nucleus) were used as a loading control. (**B**) TNF-α-induced activation of the MAPK signaling pathways in RNF4 KO MEFs. MEFs were treated with TNF-α (50 ng/mL) for the indicated periods. Cell lysates were subjected to immunoblotting with the indicated antibodies. β-actin was used as a loading control. All data are representative of at least three independent experiments.

**Figure 4 ijms-22-05796-f004:**
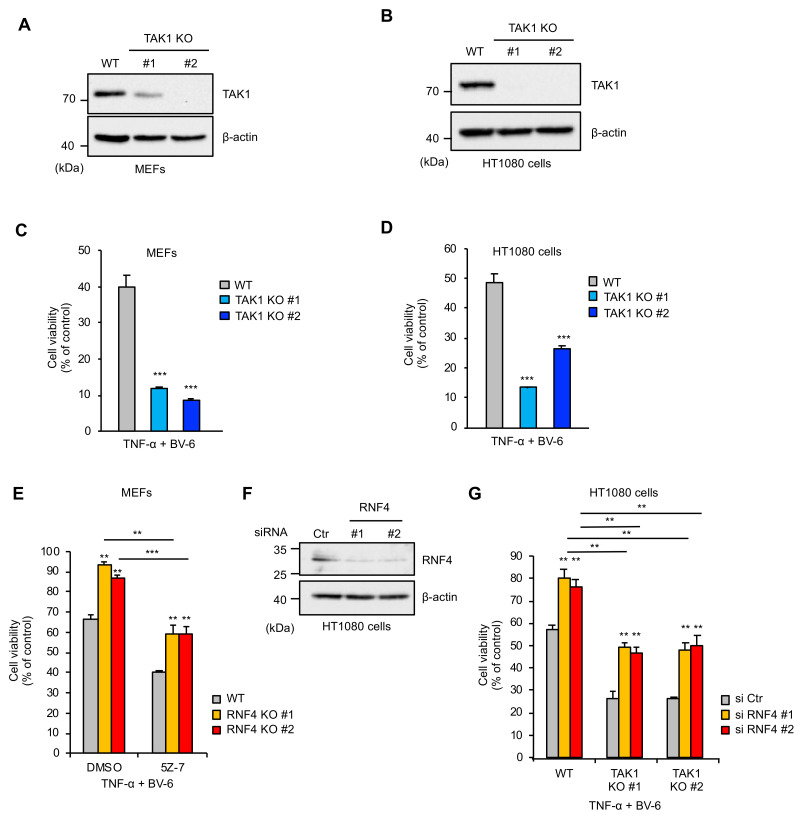
RNF4 promotes TNF-α-induced cell death independently of its inhibitory effects on the TAK1 signaling. (**A**,**B**) Immunoblot analysis of TAK1 in MEFs **(A)** and HT1080 cells (**B**). Cells were subjected to immunoblotting with the indicated antibodies. β-actin was used as a loading control. (**C**,**D**) Effect of TAK1 deletion on TNF-α-induced cell death in MEFs (**C**) and HT1080 cells (**D**). Cells were treated with TNF-α (50 ng/mL) for 12 h in the presence of BV-6 (1 μM) and then subjected to PMS/MTS assay. (**E**) Effect of 5Z-7 on TNF-α-induced cell death. MEFs were treated with TNF-α (50 ng/mL) for 12 h in the presence or absence of 5Z-7 (2 μM) and then subjected to PMS/MTS assay. (**F**) Effect of RNF4 knockdown. HT1080 cells were transfected with small interfering RNA (siRNA) for negative control or RNF4 (RNF4 #1 or RNF4 #2). After 48 h, the cells were subjected to immunoblotting with the indicated antibodies. β-actin was used as a loading control. (**G**) Requirement of RNF4 for TNF-α-induced cell death in TAK1 KO HT1080 cells. HT1080 cells were transfected with small interfering RNA (siRNA) for negative control or RNF4 (RNF4 #1 or RNF4 #2). After 48 h, the cells were treated with TNF-α (25 ng/mL) for 12 h in the presence of the BV-6 (1 μM) and then subjected to PMS/MTS assay. Data shown are the mean ± SD (n = 3). Statistical significance was tested using an unpaired Student’s *t*-test; *** *p* < 0.001, ** *p* < 0.01 (vs. control cells). All data are representative of at least three independent experiments.

**Figure 5 ijms-22-05796-f005:**
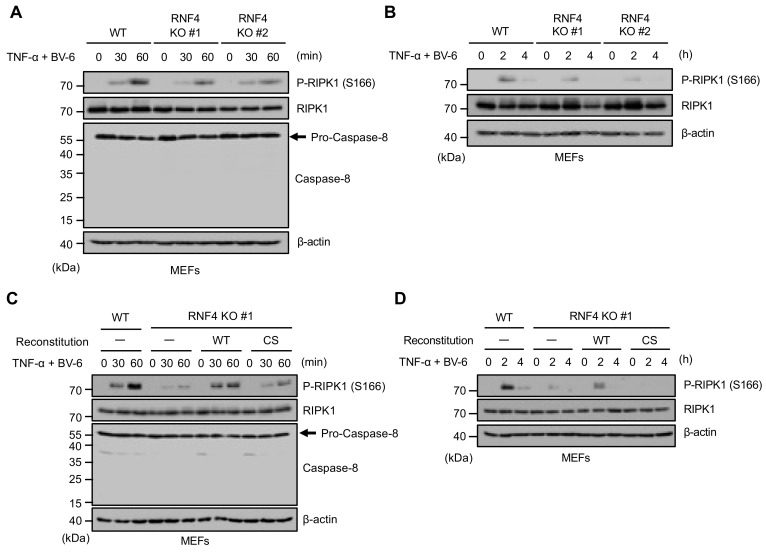
RNF4 promotes RIPK1-mediated cell death independently of its effects on the TAK1 signaling. (**A**,**B**) Autophosphorylation of RIPK1 at Ser166 triggered by TNF-α. MEFs were treated with TNF-α (50 ng/mL) for the indicated periods in the presence of BV-6 (1 μM). Cell lysates were subjected to immunoblotting with the indicated antibodies. β-actin was used as a loading control. (**C**,**D**) Effect of the RNF4 reconstitution on autophosphorylation of RIPK1 at Ser166 triggered by TNF-α. MEFs were treated with TNF-α (50 ng/mL) for the indicated periods in the presence of BV-6 (1 μM). Cell lysates were subjected to immunoblotting with the indicated antibodies. β-actin was used as a loading control. All data are representative of at least three independent experiments. (**E**) A schematic model to explain our study was described (see discussion).

## Data Availability

The data presented in this study are available in article.

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
