# Peer review of "The E3 Ubiquitin-Protein Ligase RNF4 Promotes TNF-α-Induced Cell Death Triggered by RIPK1"

_ijms, 2021, doi:10.3390/ijms22115796_

Round 1
Reviewer 1 Report
The manuscript by Shimada et al. described a role for the E3 ubiquitin ligase RNF4 in TNF-α-induced cell death mediated by a protein kinase RIPK1. They showed that TNF-α-dependent cell death via RIPK1 and its associated activation of Caspase-8 were attenuated in RNF4 KO cells. The suppression of cell death by knock-down of RNF4 was also observed in the absence of TAK1, which signaling pathway is negatively regulated by RNF4, suggesting that RNF4 promotes cell death through a mechanism independent of inhibition of the TAK1 signaling pathway. Finally, they showed that TNF-α-induced RIPK1/Ser166 phosphorylation was reduced in RNF4 KO cells.
The work is well done and convincing. Their findings that RNF4 is a positive regulator for RIPK1-mediated cell death are important, as we still have open questions about the regulation of RIPK1 activity.
Necessary improvements:
1) The authors concluded that TNF-α-induced Caspase-8 activation was more effectively rescued by the RNF4 WT expression that CS mutant (Figure 2C). However, Caspase-8 appears to be activated even in CS mutants to the same extent as in WT (especially for p43 active Caspase-8). To support the authors conclusion, other Caspase-8 activation assay (ex. Colorimetric activity assay using cell lysates) than western blot analysis would be required to quantitatively estimate the extent of Caspase-8 activation.
2) In Figure 3A, please clarify that the same amount of protein is loaded in each lane, or provide a "loading control" panel for the nuclear and cytosol fractions, respectively.
3) In Fig 4E, a band shift of RIPK1 is observed in RNF4 KO#1 (lane 4~6), whereas no such band shift is observed in Fig 4D. Please explain why such a difference occurred.
4) The authors described ULK1, TBK1 and IKKε as potential candidates of the RNF4 targets. Are the amounts of these candidates altered in the RNF4-KO cells? It would be possible to verify by western blot analysis if antibodies are available for them.
Author Response
Letter to Reviewer 1
Comment: The manuscript by Shimada et al. described a role for the E3 ubiquitin ligase RNF4 in TNF-α-induced cell death mediated by a protein kinase RIPK1. They showed that TNF-α-dependent cell death via RIPK1 and its associated activation of Caspase-8 were attenuated in RNF4 KO cells. The suppression of cell death by knock-down of RNF4 was also observed in the absence of TAK1, which signaling pathway is negatively regulated by RNF4, suggesting that RNF4 promotes cell death through a mechanism independent of inhibition of the TAK1 signaling pathway. Finally, they showed that TNF-α-induced RIPK1/Ser166 phosphorylation was reduced in RNF4 KO cells.The work is well done and convincing. Their findings that RNF4 is a positive regulator for RIPK1-mediated cell death are important, as we still have open questions about the regulation of RIPK1 activity.
Response: We would like to thank Reviewer 1 for useful advice on how to improve the manuscript. We fully agree with your advice and believe that we could fully answer to the reviewer’s requests. We hereby respectfully revise our manuscript.
Comment: 1) The authors concluded that TNF-α-induced Caspase-8 activation was more effectively rescued by the RNF4 WT expression that CS mutant (Figure 2C). However, Caspase-8 appears to be activated even in CS mutants to the same extent as in WT (especially for p43 active Caspase-8). To support the authors conclusion, other Caspase-8 activation assay (ex. Colorimetric activity assay using cell lysates) than western blot analysis would be required to quantitatively estimate the extent of Caspase-8 activation.
Response: We thank the reviewer for critical comments and advice, and we are sorry that we should have explained more clearly. The p43 fragment of caspase-8 is intermediate form that does not have the enzymatic activity, and therefore is not suitable to evaluate the caspase-8 activation. In general, the activation status of caspase-8 is evaluated by the amount of the p18 fragments of caspase-8 that possesses the enzymatic activity. As shown in Fig. 2C in the revised manuscript, a smaller amount of the p18 fragments was observed in the RNF4 CS mutants reconstituted cells when compared with the WT reconstituted cells. This data shows the requirement of E3 ligase activity of RNF4. Moreover, the colorimetric caspase-8 assay showed the significant difference between WT and CS mutants reconstituted cells (Fig. 2D in the revised manuscript). Thus, thanks to the reviewer’s advice, our claim that RNF4-induced caspase-8 activation requires its E3 ubiquitin ligase activity is consolidated. Accordingly, we modified the RESULT section by introducing following sentence: ‘Moreover, colorimetric caspase-8 assay confirmed the abrogation in RNF4 KO MEFs, showing that RNF4 acts upstream of caspase-8, and stimulates the pro-apoptotic process (Figure 1H).
’--- page2, line 93.
Comment: 2) In Figure 3A, please clarify that the same amount of protein is loaded in each lane, or provide a "loading control" panel for the nuclear and cytosol fractions, respectively.
Response: We thank the reviewer for reasonable comments. We fully agree with the reviewer’s comment, and added the"loading control" panels for both fractions (Fig. 3A in the revised manuscript).
Comment: 3) In Fig 4E, a band shift of RIPK1 is observed in RNF4 KO#1 (lane 4~6), whereas no such band shift is observed in Fig 4D. Please explain why such a difference occurred.
Response: We thank the reviewer for critical comments. As the reviewer indicated, the band shift seems to be strange. We therefore replaced the data with appropriate one (Fig. 5C in the revised manuscript).
Comment: 4) The authors described ULK1, TBK1 and IKKε as potential candidates of the RNF4 targets. Are the amounts of these candidates altered in the RNF4-KO cells? It would be possible to verify by western blot analysis if antibodies are available for them.
Response: We thank the reviewer for good advice. We fully agree with the reviewer’s comment, and detected the expression levels of these candidates expect for IKKε (an antibody did not work or IKKε does not express in HT1080 cells). As shown in the attachment file to reviewer 1, we observed reduced expression of ULK1 in RNF4 KO cells, suggesting that RNF4 regulates the expression levels of ULK1. However, the precise mechanisms by which RNF4 regulates ULK1 remains unclear. Therefore, this issue will be our future work.
Reviewer 2 Report
Dear Authors,
Your study provides novel insight of the mechanistic contribution of RNF4 in TNF signal transduction, arguing for a direct effect of its ubiquitin-ligase activity on RIPK1 phosphorylation. One may regret though that your demonstration fails to address whether the effects of RNF4 are mediated through direct regulation of RIPK1 ubiquitination. This could easily be addressed and strengthen your study.
Other comments :
Given that RIPK1 appears to be central to your demonstration, it would be nice to show its potential cleavage and phosphorylation in all western blot experiments (see Fig 1G; 2C) as well as the full blot fig 4D and E.
Along the line, could you please explain the differential RIPK1 migration profile figure 4E. Is this reproducible ?
Last Figure 2C, how do you explain that caspase-8 cleavage appears to be similar in RNF4 WT vs CS reconstituted cells, despite the fact that the effects on cell viability is different ?
Author Response
Letter to Reviewer 2
Comment: Dear Authors,
Your study provides novel insight of the mechanistic contribution of RNF4 in TNF signal transduction, arguing for a direct effect of its ubiquitin-ligase activity on RIPK1 phosphorylation.
Response: We would like to thank Reviewer 2 for useful advice on how to improve the
manuscript. We fully agree with your advice and believe that we could fully answer to
the reviewer’s requests. We hereby respectfully revise our manuscript.
Comment: One may regret though that your demonstration fails to address whether the effects of RNF4 are mediated through direct regulation of RIPK1 ubiquitination. This could easily be addressed and strengthen your study.
Response: We thank the reviewer for critical comments. The idea that RNF4 directly regulates RIPK1 ubiquitination is reasonable, and so we tested this possibility. However, we could not see the RIPK1 ubiquitination under the experimental conditions (TNF and BV6 treatment). Moreover, in this regard, a previous study has shown that RIPK1 is not ubiquitinated during proapoptotic processes (PMID: 18570872 DOI: 10.1016/j.molcel.2008.05.014). Therefore, we speculate that RIPK1 is not a direct target of RNF4. In any case, this issue will be uncovered in our future work.
Comment: Given that RIPK1 appears to be central to your demonstration, it would be nice to show its potential cleavage and phosphorylation in all western blot experiments (see Fig 1G; 2C) as well as the full blot fig 4D and E.
Response: According to the reviewer’s comment, we investigated the cleavage and phosphorylation in all western blot experiments. The RIPK1 phosphorylation at late time point like Fig. 1G was detectable, and the data are added in Fig. 5B in the revised manuscript, because we mentioned the RIPK1 phosphorylation in the chapter 5 of the RESULT section. Moreover, we added the panels of caspase-8 at early time point (0-60 min) like Fig. 4D in the original manuscript, although the cleavage of caspases was not observed in the points (Fig. 5A and 5C in the revised manuscript).
Comment: Along the line, could you please explain the differential RIPK1 migration profile figure 4E. Is this reproducible?
Response: We thank the reviewer for critical comments. As the reviewer indicated, the band shift seems to be strange. We therefore replaced the data with appropriate one (Fig. 5D in the revised manuscript).
Comment: Last Figure 2C, how do you explain that caspase-8 cleavage appears to be similar in RNF4 WT vs CS reconstituted cells, despite the fact that the effects on cell viability is different?
Response: We thank the reviewer for critical comments, and we are sorry that we should have explained more clearly. The p43 fragment of caspase-8 is intermediate form that does not have the enzymatic activity, and therefore is not suitable to evaluate the caspase-8 activation. In general, the activation status of caspase-8 is evaluated by the amount of the p18 fragments of caspase-8 that possesses the enzymatic activity. As shown in Fig. 2C in the revised manuscript, a smaller amount of the p18 fragments was observed in the RNF4 CS mutants reconstituted cells when compared with the WT reconstituted cells. This data shows the requirement of E3 ligase activity of RNF4. Moreover, to quantify the caspase-8 activation, we performed colorimetric caspase-8 assay. As shown in Fig. 2D in the revised manuscript, we found that there was significant difference between WT and CS mutants reconstituted cells. Thus, we believe that caspase-8 activation is correlated with cell death induction.
Reviewer 3 Report
The authors presented that RNF4, a E3 ligase, is required for TNF-induced RIPK1-dependent cell death through the promoting of RIPK1 phosphorylation. Although the finding is interesting and is potentially important in the field of cell death, authors should supply more information on some data for readers.
I have following observations
In Fig.1E and G,.
While KO#2 clone showed more resist to death compare to KO#1, but more cleaved caspase-8 and cleaved caspase-3 are shown in KO#2. Please give an explanation to account for this data
Figure 2C.
In CS reconstituted KO cells, the amount of cleaved caspase-8 (p18) is comparable to WT cells at 6h, however the cells are still resist to cell death induced by TNF+BV6.
Please explain how does it possible? Can it be a caspase-8 independent death?
Moreover, the initial cleavage of casp8 (P48) was also not affected in CS reconstituted KO cells. And authors claim that RNF4 might target RIPK1.
Please explain by which signal induced caspase-8 cleavage in this cells under blocking of RIPK1 phosphorylation.
“2.3. RNF4 suppresses TNF-α-induced activation of the NF-κB and MAPK signaling pathways”
I am not sure one can say that MAPK signaling is enhanced by KO of RNF4 from the data of Figure 3B.
The loading amount looks not even in each lane. p38 is just one of MAPKs and it is not proper to use MAPK signaling pathway in subtitle.
In Fig. 4E.
The RIPK1 in RNF4 KO#1 cells looks modified (up-shift).
In addition, the level of RIPK1 in the KO1 cells was not changed, while the level of RIPK1 is decreased in Wt, WT or CS-reconstituted MEFs by the treatment of TNF + BV6.
And, the RIPK1 level of KO cell at time 0 is lower that of Wt or reconstituted KO cells. Is it true?
Please give a comprehensive explanation.
As the authors mentioned, there are several possible targets including RIPK1 for RNF4.
The paper would be significantly improved with the addition of data whether RNF4 directly interact with RIPK1 upon stimulation with TNF.
Miner point
In the legends of Figure 1. (C), (D), (E), and (F) should be positioned before MEFs.
I think Fig. 3C-F should be combine with Figure 4.
Figure 4D and E. Authors used a different time frame. Are there any reasons?
Author Response
Letter to Reviewer 3
Comment: The authors presented that RNF4, a E3 ligase, is required for TNF-induced RIPK1-dependent cell death through the promoting of RIPK1 phosphorylation. Although the finding is interesting and is potentially important in the field of cell death, authors should supply more information on some data for readers. I have following observations.
Response: We would like to thank Reviewer 3 for useful advice on how to improve the manuscript. We fully agree with your advice and believe that we could fully answer to the reviewer’s requests. We hereby respectfully revise our manuscript.
Comment: In Fig.1E and G. While KO#2 clone showed more resist to death compare to KO#1, but more cleaved caspase-8 and cleaved caspase-3 are shown in KO#2. Please give an explanation to account for this data.
Response: We thank the reviewer for critical comments. We fully agree with the reviewer’s comment, and repeated the experiments to explain the discrepancy. As shown in Fig. 1E and 1G in the revised manuscript, we found that there is a good correlation between the caspase activation and cell death.
Comment: Figure 2C. In CS reconstituted KO cells, the amount of cleaved caspase-8 (p18) is comparable to WT cells at 6h, however the cells are still resist to cell death induced by TNF+BV6. Please explain how does it possible? Can it be a caspase-8 independent death?
Response: We thank the reviewer for reasonable comments. As you pointed out, the difference between WT and CS mutant reconstituted cells is small. Therefore, to quantify the difference, we performed colorimetric caspase-8 assay. As shown in Fig. 2D in the revised manuscript, we found that there was significant difference between WT and CS mutants reconstituted cells. Thus, we believe that caspase-8 activation is correlated with cell death induction.
Comment: Moreover, the initial cleavage of casp8 (P43) was also not affected in CS reconstituted KO cells. And authors claim that RNF4 might target RIPK1. Please explain by which signal induced caspase-8 cleavage in this cells under blocking of RIPK1 phosphorylation.
Response: We thank the reviewer for incisive comments. Although the observation that the initial cleavage of casp8 was proceeded without the RIPK1 phosphorylation is interesting, there is not a convincing explanation because the functional roles of the RIPK1 phosphorylation in the caspase-8 activation are still unclear. Our data imply the possibility that phosphorylated RIPK1 promotes secondary cleavage of caspase-8. In any case, the colorimetric caspase-8 assay shown in Fig. 2D in the revised manuscript suggests that E3 ubiquitin ligase activity of RNF4 is required for the caspase-8 activation.
Comment: “2.3. RNF4 suppresses TNF-α-induced activation of the NF-κB and MAPK signaling pathways” I am not sure one can say that MAPK signaling is enhanced by KO of RNF4 from the data of Figure 3B. The loading amount looks not even in each lane. p38 is just one of MAPKs and it is not proper to use MAPK signaling pathway in subtitle.
Response: We thank the reviewer for reasonable comments. According to the reviewer’s advice, we repeated the experiments and replaced the data with appropriate one (Fig. 3B in the revised manuscript). Moreover, we added the ERK and JNK blots, because they are representative MAP kinases as well as p38.
Comment: In Fig. 4E.
The RIPK1 in RNF4 KO#1 cells looks modified (up-shift). In addition, the level of RIPK1 in the KO1 cells was not changed, while the level of RIPK1 is decreased in Wt, WT or CS-reconstituted MEFs by the treatment of TNF + BV6. And, the RIPK1 level of KO cell at time 0 is lower that of Wt or reconstituted KO cells. Is it true? Please give a comprehensive explanation.
Response: We thank the reviewer for critical comments. As the reviewer indicated, the band shift seems to be strange. We therefore replaced the data with appropriate one (Fig. 5B in the revised manuscript).
Comment: As the authors mentioned, there are several possible targets including RIPK1 for RNF4. The paper would be significantly improved with the addition of data whether RNF4 directly interact with RIPK1 upon stimulation with TNF.
Response: We thank the reviewer for critical comments. The idea that RNF4 directly interacts with RIPK1 is reasonable, and so we tested this possibility. However, we could not see any interaction between RNF4 and RIPK1, even if both were overexpressed. Therefore, we speculate that RIPK1 indirectly regulates the RIPK1 activity.
Miner point
Comment: In the legends of Figure 1. (C), (D), (E), and (F) should be positioned before MEFs.
Response: We thank and fully agree with the reviewer’s comment, and modified them according to the advice.
Comment: I think Fig. 3C-F should be combine with Figure 4.
Response: We thank the reviewer for critical comments. We fully agree with the reviewer’s idea, and combined them.
Comment: Figure 4D and E. Authors used a different time frame. Are there any reasons?
Response: We thank the reviewer for critical comments. In this regard, there is no reason. Therefore, we showed the data with the same time course (Fig. 5A and 5C in the revised manuscript).
Round 2
Reviewer 3 Report
I think the authors have included new and better figiures and these changes have improved the overall quality of paper.